# ZeRO++: Extremely Efficient Collective Communication for Large Model Training

**Guanhua Wang**[*1], **Heyang Qin**[*1], **Sam Ade Jacobs**[1], **Xiaoxia Wu**[1], **Connor Holmes**[2],
**Zhewei Yao**[3], **Samyam Rajbhandari**[3], **Olatunji Ruwase**[1], **Feng Yan**[4], **Lei Yang**[5], **Yuxiong He**[3]

Microsoft DeepSpeed[1],   OpenAI[2],   Snowflake[3],
University of Houston[4],   University of Nevada, Reno[5]

## Abstract

Zero Redundancy Optimizer (ZeRO) has been used to train a wide range of large language models on massive GPU clusters due to its ease of use, efficiency, and good scalability. However, when training on low-bandwidth clusters, and/or when small batch size per GPU is used, ZeRO's effective throughput is limited by communication overheads. To alleviate this limitation, this paper introduces ZeRO++ composing of three communication volume reduction techniques (low-precision all-gather, data remapping, and low-precision gradient averaging) to significantly reduce the communication volume up to 4x that enables up to 2.16x better throughput at 384 GPU scale. Our results also show ZeRO++ can speedup the RLHF training by 3.3x compared to vanilla ZeRO. To verify the convergence of ZeRO++, we test up to 13B model for pretraining with 8/6-bits all gather and up to 30B model for finetuning with 4/2-bits all gather, and demonstrate on-par accuracy as original ZeRO (aka standard training). As a byproduct, the model trained with ZeRO++ is naturally weight-quantized, which can be directly used for inference without post-training quantization or quantization-aware training.

## 1 Introduction

The size of deep learning (DL) models has increased from 100 million to over 500+ billion parameters, ranging from BERT (Devlin et al., 2018) to Megatron-Turing NLG (Smith et al., 2022). With the increase in model size, the memory and compute requirements for training have increased significantly beyond the capability of a single accelerator (e.g., a GPU). Training these massive models requires the efficient aggregation of computing power and memory across hundreds or even thousands of GPU devices. There are two popular approaches to alleviate this, namely 3D parallelism (Narayanan et al., 2021; Team & Majumder, 2020) and Zero Redundancy Optimizer (ZeRO) (Rajbhandari et al., 2020).

Compared to 3D parallelism, ZeRO is easier to use without model code refactoring. ZeRO is a memory efficient variation of data parallelism (Ben-Nun & Hoefler, 2019; Dean et al., 2012) where model states are partitioned across all the GPUs, instead of being replicated, and reconstructed using gather based communication collectives on-the-fly during training. This allows ZeRO to effectively leverage the aggregate GPU memory across machines, at the expense of minimal communication overhead (50%) compared to standard data parallel training (2M vs 3M for model size of M) (Rajbhandari et al., 2020), while still achieving excellent throughput scalability (Rajbhandari et al., 2021).

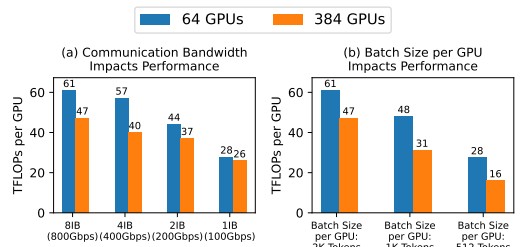

Figure 1: Training throughput are constrained by network bandwidth and batch size per GPU.

However, the communication overhead of ZeRO can limit throughput in two important scenarios (1) low-bandwidth cluster and (2) low-compute training scenario, e.g., the batch size per GPU is

---

* equal contribution

small. We demonstrate these two cases in Figure 1. As can be seen, as the bandwidth of the cluster becomes smaller and/or the training batch size decreases, the training efficiency of ZeRO significantly reduces.

To overcome the communication overhead of ZeRO, we here present a novel system of communication optimizations collectively called ZeRO++, for which the contribution can be summarized:

**Quantized Weight Communication for ZeRO (qwZ).** First, in order to reduce parameter communication volume during forward all-gather (which is M for ZeRO), we adopt quantization on weights to shrink down each model parameter from FP16 to lower-precision data type (e.g., 8/6-bits) before communicating. To preserve decent model training precision, we adopt block-based quantization (Dettmers et al., 2022; Yao et al., 2022), which conducts independent quantization on each subset of model parameters. There is no existing implementation for high performance block-based quantization. Thus, we implement highly optimized quantization CUDA kernels from scratch. Another great byproduct of qwZ is that it automatically converts the model weight from half-precision (16-bits) to lower precision for efficient inference without any inference quantization methods. Particularly, as compared to the current popular post-training quantization methods (Frantar et al., 2022), qwZ can push the weight-precision to 2-bits for finetuned model without significant accuracy drop.

**Hierarchical Weight Partition for ZeRO (hpZ).** Second, to reduce the communication overhead of all-gather on weights during backward (which is $M$ for ZeRO), we trade GPU memory for communication. More specifically, instead of spreading whole model weights across all the machines, we maintain a full model copy within each node. At the expense of higher memory overhead, this allows us to replace the expensive cross-machine all-gather on weights with intra-machine all-gather, which is substantially faster due to much higher intra-machine communication bandwidth. hqZ reduces the all-gather for backward from volume $M$ to 0.

**Quantized Gradient Communication for ZeRO (qgZ).** Third, reducing the communication cost of gradients using reduce-scatter (which is $M$ for ZeRO) is even more challenging. Directly applying quantization to reduce communication volume is infeasible due to the training accuracy drop. To better preserve the accuracy, we propose a novel hierarchical all-to-all based gradient scatter-reduction schedule, where an intra-node gradient reduction is first applied and then a inter-node quantized gradient communication (e.g., 8/4-bits) is used resulting in 2/4x communication volume reduction. To achieve the best outcome, we incorporate pipelining intra-node and inter-node communication and conducting CUDA kernel fusion.

By incorporating all three components above, we reduce the cross-node communication volume by **4x** from $3M$ down to less than $0.75M$ ($< 0.5M$ for forward, 0 for backward, and $0.25M$ for gradient reduce-scatter). We extensively test ZeRO++'s system performance, and show (i) scalability of GPT-3 like models on up to 384 GPUs achieving over 45% of sustained peak throughput, (ii) consistent speedup of up to 2.4x over ZeRO across models ranging from 10-138B parameters, (iii) comparing with baseline in 4x higher bandwidth cluster, ZeRO++ achieve better throughput in low-bandwidth setting, (iv) 3.3x better throughput for RLHF training and generation.

To verify the convergence of ZeRO++, we tested it on both pretraining and fine-tuning. Our results show that ZeRO++ can (1) achieve on-par training loss as ZeRO for up to 13B GPT-3 models on pretraining with both 8-bits and 6-bits qwZ, and (2) realize similar (slightly worse) model quality as ZeRO for OPT-30B finetuning with 4-bits (2-bits) qwZ. The 2-bits qwZ trained model can be directly served for inference, which is significantly better than current post-training quantization methods (PTQ) (Frantar et al., 2022) while ZeRO++ even saves training cost and PTQ cost.

## 2 BACKGROUND AND RELATED WORK

This section only describes the core background needed for understanding the techniques discussed in this paper. A broader discussion of the related work is provided in Appendix A .

### 2.1 ZERO OPTIMIZER

ZeRO is a memory-optimized solution for data parallel training. ZeRO partitions and distributes all model states (i.e., parameters, gradients, optimizer states) among GPUs in use and recollects model states only when the layer needs to be computed. There are three different stages for using ZeRO

to optimize on-device memory usage. In ZeRO stage 1 (ZeRO-1), only optimizer states are split and spread across all GPUs in use. ZeRO stage 2 (ZeRO-2) partitions both optimizer states and gradients, and ZeRO stage 3 (ZeRO-3) splits all three components of model states as parameters, gradients, and optimizer states.

ZeRO-3 is the most memory efficient solution for model training at large scale, but at the cost of more collective communications. Algorithm 1 illustrates the high-level pseudocode for ZeRO-3. During model training, ZeRO-3 lazily schedules the fetching of parameters until the computation needs to happen on a particular layer. Before forward propagation, ZeRO launches an all-gather to collect the full model weights and then computes the forward pass (line 2-3) of Algorithm 1. Then ZeRO empties the all-gather weights buffer after forward

---

**Algorithm 1:** ZeRO algorithm

**Input** : $model, worldSize$
**Output:** $model$
1 **while** $model\ not\ converged$ **do**
2      $all\_gather\_Parameters(worldSize)$;
3      $model.forward()$;
4      $partition(worldSize)$;
5      $all\_gather\_Parameters(worldSize)$;
6      $model.backward()$;
7      $partition(worldSize)$;
8      $reduce\_scatter\_Gradients(worldSize)$;
9      $optimizer.step()$;
10 **end while**
11 **Return:** $model$

---

computation completes (line 4). During backward, ZeRO re-collects all model weights again via a second all-gather (line 5) to calculate gradients (line 6). Once gradients are calculated on each GPU, ZeRO empties weights buffer again (line 7) and conducts a reduce-scatter operation to do gradient averaging and re-distribution (line 8). Model states and parameters are updated in the optimizer step (line 9).

In summary, to minimize the on-device memory footprint using ZeRO-3, at each iteration there are three collective communication operations: two all-gather on weights and one reduce-scatter on gradients.

## 2.2 Communication Reduction Techniques

**Quantization** is often used to reduce memory footprint, and data movement volume by using low precision to represent data (Dettmers, 2015; Dettmers et al., 2022). However, the loss of information from representing high precision data with lower precision often comes with accuracy degradation. Related work aims to enhance quantization accuracy by addressing the challenges associated with differences in number ranges and granularity between high and low precision data. Some related work (Zhao et al., 2019) propose to filter the outliers in data to mitigate the gap in numerical ranges. Yet their accuracy hinges on the quality of outlier filtering and it brings extra filtering overhead. Dettmers et al. (2022) propose to use block based quantization on optimizer states to improve the quantization accuracy. However, it requires modifications to the model structure, thereby limiting its usability.

**Gradient Compression** techniques, including 1-bit SGD, 1-bit Adam, and 1-bit Lamb, have been developed to optimize gradient communication in distributed training by utilizing minimal bit representation (Seide et al., 2014; Tang et al., 2021; Li et al., 2021). However, direct application of these methods to ZeRO-3 is infeasible as they assume a full view of optimizer states (OS) across GPUs, which is not the case in ZeRO-3.

**ZeRO Communication Reduction.** Recent optimizations on ZeRO-3, like MiCS (Zhang et al., 2022c) and HSDP (Zhang et al., 2022a), divide the GPU cluster into sub-groups and leverage high bandwidth intra-node interconnect, or hierarchical communication to minimize communication volume. $hpZ$ in ZeRO++ adopts a similar approach yet it performs only secondary partitioning on weights, while keeping all other model states partitioned across all GPUs. This allows hpZ to achieve significant communication reduction without the massive memory overhead of MiCS.

## 3 Design

In this section, we elaborate on the design of our three key optimizations in ZeRO++ introduced in Section 1 for reducing the communication overhead of ZeRO: i) Quantized Weight Communication for ZeRO ($qwZ$), ii) Hierarchical Partitioning for ZeRO ($hpZ$), and iii) Quantized Gradient communication for ZeRO ($qgZ$). We further discuss the end-to-end impact of these optimizations to reduce to total communication volume of ZeRO in Appendix B.4.

## 3.1 Quantized Weight Communication for ZeRO ($qwZ$)

As discussed in Section 2.1, ZeRO partitions the model weights across all the ranks (i.e., GPUs) and fetches the FP16 weights layer-by-layer right before they are needed in computation via all-gather for the forward and backward of each training iteration. To reduce the communication overhead of forward all-gather on weights, $qwZ$, quantizes FP16 weights to lower precision right during the all-gather, and dequantizes them back to FP16 on the receiver side, and then conducts layer computation.

While this reduces the communication volume of the all-gather by more than 2x, doing so naively results in two major issues: i) the lowering of precision results in significant accuracy degradation during training as discussed in Section 2.2 , and ii) the quantization and dequantization overhead negates any throughput gain from communication volume reduction.

To improve quantization accuracy we use blocksize based quantization (Yao et al., 2022; Shen et al., 2020) (further details in Appendix B.1). To mitigate the quantization and dequantization overhead, we develop custom optimized implementation of $qwZ$ (further details in Appendix C)

**Additional Benefits of $qwZ$**

i) *Automatic Quantization $qwZ$* automates parameter quantization during training, allowing higher compression (to 2 bits) with minimal accuracy loss, thereby obviating the need for post training quantization (Frantar et al., 2022; Yao et al., 2022) (see Section 4 for more details),

ii) *Memory footprint reduction* For untrainable/frozen weights (e.g., in LoRA or multimodal training), $qwZ$ alleviates the necessity of persistent FP16 storage, which reduces memory footprint. This reduction can allow for a) training with larger batch sizes for better throughput, b) inference or training with fewer resources without running out of memory, and c) speed up inference by reducing the amount of parameter data that needs to be read from memory.

In fact, given the automatic quantization and memory footprint reduction, $qwZ$ has a similar effect for fine-tuning as QLoRA (Dettmers et al., 2023), in terms of weight quantization and memory requirement reduction, but in addition ZeRO++ with $qwZ$ can support significantly larger model sizes than QLoRA due to the weight partitioning from ZeRO.

## 3.2 Hierarchical Partitioning for ZeRO ($hpZ$)

With $hpZ$, we eliminate the inter-node all-gather during the backward pass by holding secondary FP16 weights partition within each node. We do this by creating a hierarchical partitioning strategy consisting of two partitions: first, all model states are partitioned globally across all devices as in ZeRO-3, which we call primary partition. Second, a secondary copy of FP16 parameters is partitioned at the sub-global level (e.g., compute node, see Figure 2), which we call secondary partition. This secondary copy of FP16 parameters is replicated across multiple secondary partitions.

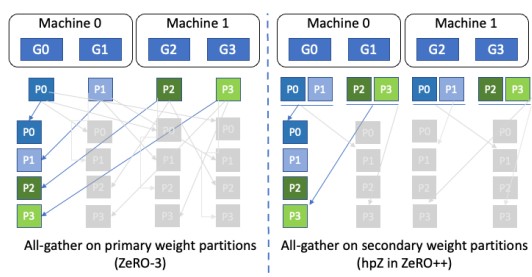

Figure 2: hpZ removes cross node traffic in backward all-gather by holding secondary weight partitions in on-device memory.

Consider a 64-node cluster, each node with 8 GPUs. Model weights are partitioned in two stages: i) across all 512 GPUs that we call primary partition, and ii) the same weights are also partitioned within a compute node across 8 GPUs, that we call secondary partition. In this example, for the secondary partition, each compute node in the cluster holds a full replica of FP16 weights partitioned among the 8 GPUs within the node, and there are 64 of such replicas in total.

**A training iteration with hpZ.** During the forward pass of a training iteration, we all-gather weights based on the primary partition across all GPUs. However, once the weights have been used during the forward pass, these weights are then partitioned based on the secondary partition. Given the temporal consistency of model parameters between forward and backward passes, when the weights are needed again during the backward pass, we all-gather weights based on this secondary group.

When the secondary partitioning is set to be inside a compute node, this avoids any inter-node communication for this all-gather. Finally, at the end of the iteration, during the optimizer step, all the model states, and the primary copy of the fp16 parameters are updated based on the primary partition. hpZ makes two changes to baseline ZeRO pseudocode in Algorithm 1: i) in line 4, parameter partitioning is based on *secondary group size*, ii) parameter all-gather preceding backward pass in line 5 is also based on *secondary group size*.

Our design of $hpZ$ can flexibly support any *secondary group size*. The group size controls how many ranks (i.e., GPUs) are in the secondary partition. It is also a measure of the memory-communication trade-off of $hpZ$ discussed in Appendix B.2.

### 3.3 Quantized Gradients Communication for ZeRO ($qgZ$)

qgZ is a novel quantized reduce-scatter algorithm based on all-to-all collectives that enables a 4x communication volume reduction of gradient reduce-scatter by replacing FP16 with INT4. qgZ has three components: 1) all-to-all-based quantized gradient reduce-scatter, 2) reducing communication volume with hierarchical collectives (details in Appendix B.3), 3) tensor slice reordering for correct gradient placement.

#### 3.3.1 All-to-all based implementation

A naive way to quantized reduce-scatter without precision loss is to apply quantization and dequantization to a ring-based reduce-scatter directly as shown on the left of Figure 3. We inject quantization and dequantization on each GPU. Once a GPU receives gradients from its predecessor, we dequantize it to full precision and conduct a local reduction. After that we quantize local reduction output and pass it to its successor. To complete the whole reduce-scatter, the number of sequential quantization and dequantization kernels is equal to the number of GPUs in use. Thus, applying quantization on existing ring based reduce-scatter collective will lead to high communication latency and low value precision due to multiple sequential quantization and dequantization steps. Although recent tree-based collectives like Blink(Wang et al., 2020) could reduce the number of sequential kernels from n to log(n), the long latency and low precision issue is not completely resolved.

To overcome this, we completely abandon existing ring-based reduce-scatter approach and incorporate 1-hop all-to-all collective for our gradient communication. As shown on the right of Figure 3, we first apply quantization on a given tensor, then we conduct all-to-all communication among all the GPUs. After all-to-all, we apply another dequantization to recover the data precision and then reduce on high-precision values to get the final gradient reduction output. By replacing ring-based solution with our all-to-all collective, we reduce the number of sequential quantization+dequantization kernels from the number

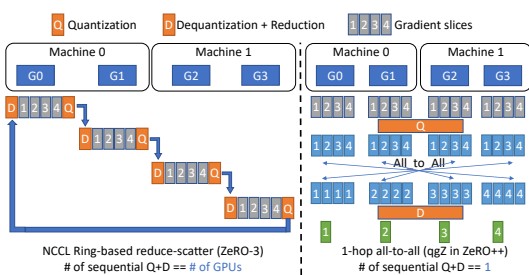

Figure 3: Comparison between ZeRO-3 ring-based reduce-scatter and qgZ 1-hop all-to-all.

of GPUs to 1. We further reduce cross node communication volume by incorporating hierarchical collectives named *2-hop all-to-all*, which is detailed in Appendix B.3.

#### 3.3.2 Tensor slice reordering for correct data placement

With our *2-hop all-to-all*, inter-node communication volume is as expected, however, this introduces a gradient misplacement issue. We describe this issue using a 2x2 example, where we have 2 machines and each machine has 2 GPUs. As shown in Figure 4, the correct final gradient placement is shown as green boxes in the figure, where GPU 0 holds final gradient partition 1, GPU 1 holds gradient partition 2, so on and so forth.

Our 2-step all-to-all communication works as follows, first we divide all gradients on each GPU into 4 chunks, then conduct our intra-node all-to-all. After intra-node all-to-all finishes, GPU0 (i.e., G0) holds partial aggregated gradient partitions 1,2 whereas G1 holds gradient partitions 3,4. Same thing happens on G2 and G3. Since G1 does not have gradient partition 2 (which is supposed to be

Figure 4: Gradient partition misplacement when applying hierarchical all-to-all in qgZ.

Figure 5: Tensor slices reordering to correct gradient misplacement in qgZ.

held by G1) while G2 does not have gradient partition 3, after inter-node all-to-all, there is gradient misplacement issue on both G1 and G2.

We address this with tensor slice reordering. In Figure 5, before intra-node all-to-all, we first swap tensor slice order of slice 2 and 3, which is shown as orange arrows. Then after intra-node all-to-all is completed, G1 now has gradient 2 while G2 has gradient 3. Therefore, after inter-node all-to-all, all GPUs get the correct gradient placement. Mathematically, given X GPUs per node and Y nodes in total, each GPU holds X*Y gradient slices initially. Tensor slice reordering works as follows:

$$before : [0, 1, 2, 3, 4, ...YX - 3, YX - 2, YX - 1] \qquad (1)$$

$$after : [0, X, 2X, ...(Y - 1)X, 1, X + 1, (Y - 1)X + 1, ...YX - 1] \qquad (2)$$

Based on Eq. 1 and 2, we can map each original tensor slice position (i.e., Eq. 1) to new tensor slice position (i.e., Eq. 2) on each GPU to correct final gradient misplacement issue.

## 3.4 Optimized Implementation

To optimize ZeRO++, we implemented key optimizations: i) overlapping compute streams and, ii) optimizing CUDA kernels. Overlapping involves concurrent execution of quantization computation and communication during weight all-gathering, leveraging asynchronous quantization and execution order tracking. The CUDA kernels were optimized for quantization, dequantization, and tensor slice reordering to fully utilize device memory bandwidth and minimize memory traffic by 9x in $qgZ$. Additionally, a hierarchical approach in gradient communication and a generalized tensor slice reordering scheme are introduced to minimize latency and address the needs of different pipeline stages. These optimizations allow ZeRO++ to efficiently leverage 4x communication volume reduction, enhancing its throughput. Please refer to the Appendix C for implementation details and communication analysis.

## 4 Evaluation

This section evaluates ZeRO++ in three areas. First, it shows end-to-end throughput scalability and speedup over baseline for standard and RLHF training across different models, model sizes, hardware configurations and cluster settings, demonstrating consistent speedup (up to 3.3x) across the board. Second, it shows convergence properties of ZeRO++ for both pre-training and fine-tuning demonstrating its robustness and tolerance to extreme quantization all the way down to 2-bits. Third, it shows ablation studies demonstrating the impact of each component of ZeRO++ and the effectiveness of our kernel optimizations.

## 4.1 Methodology

**Hardware:** 24 NVIDIA DGX-2 nodes where each with 16 V100 SXM3 32 GB GPUs. The nodes are connected by InfiniBand (IB) with NVIDIA SHARP support, achieving total inter-node bandwidth of over 800 Gbps. To evaluate ZeRO++ in clusters under different network environments, we show the performance of ZeRO++ running with different cross-node bandwidths by enabling from 1 to 8 IB connections (i.e., 100 Gbps to 800 Gbps).

**Baseline:** We use ZeRO-3 as the baseline given its ease-to-use for training giant models at large scale. To evaluate the performance of our optimized kernels, we also implemented ZeRO++ with Py-Torch quantization and non-fused kernels as baselines for our ablation study.

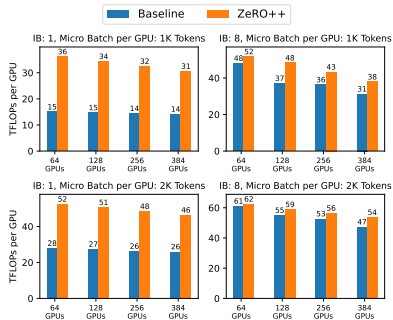

Figure 6: Scalability on up to 384 GPUs of 18B model with different numbers of InfiniBand connections and tokens per GPU.

Table 1: End-to-end speedup of ZeRO++ on V100 and A100 GPUs.

**(a) 384 V100 GPUs**

| Model Size | Tokens per GPU | 1 IB Connection | | | 8 IB Connections | | |
|---|---|---|---|---|---|---|---|
| | | Baseline TFLOPs | ZeRO++ TFLOPs | Speedup | Baseline TFLOPs | ZeRO++ TFLOPs | Speedup |
| 138B | 2K | 19.96 | 37.90 | 90% | 47.55 | 55.30 | 16% |
| 138B | 1K | 11.25 | 21.81 | 94% | 34.19 | 44.38 | 30% |
| 91B | 2K | 19.99 | 38.06 | 90% | 47.74 | 56.26 | 18% |
| 91B | 1K | 11.27 | 21.93 | 95% | 34.49 | 44.36 | 29% |
| 49B | 2K | 20.06 | 38.08 | 90% | 48.05 | 56.24 | 17% |
| 49B | 1K | 11.27 | 21.95 | 95% | 34.54 | 44.46 | 29% |
| 18B | 2K | 25.98 | 46.40 | 79% | 47.31 | 53.65 | 13% |
| 18B | 1K | 14.15 | 30.57 | 116% | 31.27 | 37.87 | 21% |

**(b) 32 A100 GPUs**

| Model Size | Tokens per GPU | 1 IB Connection | | | 4 IB Connections | | |
|---|---|---|---|---|---|---|---|
| | | Baseline TFLOPs | ZeRO++ TFLOPs | Speedup | Baseline TFLOPs | ZeRO++ TFLOPs | Speedup |
| 18B | 2K | 64.99 | 111.66 | 71.8% | 134.65 | 134.94 | 0.2% |
| 18B | 1K | 34.52 | 65.2 | 88.8% | 85.16 | 93.12 | 9.3% |

**Model Configurations:** We use transformer models for evaluation including GPT, OPT, andLLaMA for various model sizes. Given Megatron-Turing-NLG (Smith et al., 2022) training 530B model on 2K GPUs using 2K tokens per GPU (i.e., micro batch size), we evaluate ZeRO++ with the same 2k tokens per GPU setting. We also evaluate on 1K tokens per GPU to test ZeRO++ with more extreme scale scenario. The number of layers and hidden sizes are adjusted to have models of different sizes. Please refer to the appendix and our open-sourced evaluation scripts for hyperparameters and other training details.

## 4.2 E2E SYSTEM EVALUATIONS

We evaluate ZeRO++ end-to-end performance and present an ablation study here. One key metric we use here is the percentage of *peak performance*, which is shown as Eq. 3.

$$\text{peak\_performance} = \text{achieved\_TFLOPs}/\text{max\_TFLOPs} \qquad (3)$$

E.g. when we use V100 GPU, its max_TFLOPs is 120 TFLOPs (NVIDIA V100 datasheet) for mixed precision computation. Thus, our reported *peak performance* refers to the percentage number of achieved_TFLOPs/120.

**Scalability upto 384 GPUs** In Figure 6, we present ZeRO++ scalability evaluation from 64 to 384 GPUs with 18B model on both low (1 IB) and high (8 IB) bandwidth clusters. On low bandwidth cluster, ZeRO++ achieves 30% and 38.3% of peak performance (120 TFLOPs) even at 384 GPUs for 1K and 2K batch sizes, which is much higher compared to 12.5% and 21.6% as baseline peak performance. This presents up to **2.4x** better throughput. On high bandwidth cluster, despite having significantly more bandwidth, ZeRO++ still enables up to 1.29x better throughput, and can achieve up 45% of sustained peak throughput at 384 GPUs. ZeRO++ significantly speed up large scale training for low bandwidth clusters while archiving decent speedup even on high bandwidth clusters.

**Throughput for different model sizes and GPU architectures** Table 1(a) compares training throughput for models of 18B-138B on 384 GPUs between ZeRO++ and baseline on both low and high bandwidth clusters. On low bandwidth cluster, ZeRO++ consistently achieves over 31.5% and 18.1% peak performance for 2K and 1K batch sizes on all models. Compared with the baseline peak performance of 16.6% and 9.3%, the speedup is up to **2.16x**. On high bandwidth cluster, ZeRO++ peak performances are 44.7% and 31.5%, which is 1.3x over the baseline peak performance of 31.5% and 26.0%.

Table 1(b) illustrates the performance of ZeRO++ on 32 A100 GPUs, demonstrating that ZeRO++ surpasses the baseline even on a smaller scale cluster. Specifically, in low-bandwidth clusters, ZeRO++ exceeds the baseline by 88.8%, and in high-bandwidth clusters, it achieves a 9.3% improvement over the baseline. ZeRO++ exhibits robustness and uniform speedup across varying model and batch sizes, as well as differing network bandwidths and GPU architectures.

**Democratization for large scale training** Figure 7 compares the throughput of ZeRO++ on a low cross-node bandwidth (200 Gbps as 2 IB) cluster with the baseline running on 800 Gbps high-bandwidth (8 IB) cluster. For a small model of 18B, ZeRO++ achieves a higher peak performance of

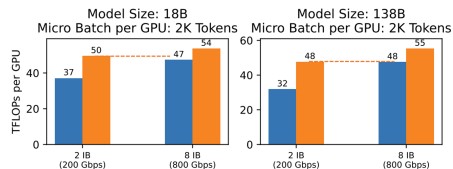

Figure 7: ZeRO++ achieves high bandwidth cluster performance with much lower bandwidth.

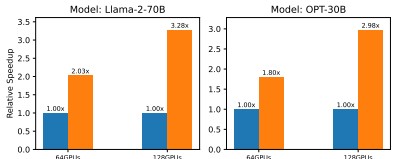

Figure 8: ZeRO++ achieves up to 3.3x better throughput for RLHF training.

Table 2: Perplexity after fine-tuning different models with various quantization bits in ZeRO++.

|  | OPT-1.3B | OPT-13B | LLaMA-30B |
|---|---|---|---|
| FP16 (Baseline) | 1.804 | 1.698 | 1.490 |
| ZeRO++ 6-bits | 1.809 | 1.705 | 1.500 |
| ZeRO++ 4-bits | 1.830 | 1.705 | 1.494 |
| ZeRO++ 2-bits | 2.218 | 1.809 | 1.544 |

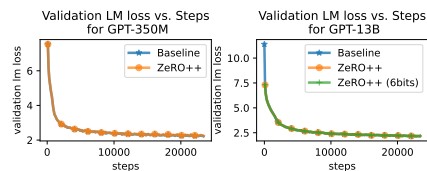

Figure 9: Convergence curves: Pretraining GPT-350M and GPT-13B on the Pile dataset.

41.6% compared with baseline peak performance of 39.1% despite running with 4x lower cross-node bandwidth. For large model of 138B, ZeRO++ and baseline achieve the same peak performance of 40%, but baseline runs at 4x higher cross-node bandwidth. This evaluation shows that ZeRO++ makes large scale training more accessible by significantly decreasing the minimum cross-node bandwidth requirement for efficient training. Furthermore, it demonstrates that optimized ZeRO++ implementation effectively translates the 4x communication reduction of ZeRO++ into real end-to-end system throughput gain.

**Speedup for RLHF** Reinforcement Learning from Human Feedback (RLHF) is a unique and commonly employed scenario in LLM training. In Figure 8, we present a comparison between ZeRO++ and the baseline for OPT-30B and LLaMA-2-70B, demonstrating that ZeRO++ can achieve up to 3.3x and 2.97x improved throughput forLLaMA-2-70B and OPT-30B respectively. The throughput is measured at step3 of RLHF training where there are both training and generation. It's noteworthy that the results were obtained with 8 InfiniBand connections on V100 GPUs, and we anticipate an even larger gap in lower bandwidth clusters. These findings underscore the versatility and efficacy of ZeRO++ across diverse training scenarios and various model families.

## 4.3 MODEL CONVERGENCE ANALYSIS

This section assesses ZeRO++'s impact on the convergence of large models during both pretraining and fine-tuning stages.

**Pretraining.** We analyze the pretraining of GPT-350M and GPT-13B models on the Pile dataset (Biderman et al., 2022), employing ZeRO++ with non-blocked quantization, ZeRO++ (with blocked quantization), and ZeRO-3 as the baseline. To maintain fairness, all hyperparameters remain consistent between baseline and ZeRO++ trainings. Convergence is measured by the validation LM loss.

Figure 9 displays the comprehensive pretraining trace. Training with 8-bit non-blocked quantization diverged initially, rendering no visible data. Conversely, ZeRO++ with 8-bit blocked quantization aligns closely with the baseline, reinforcing our prior analysis that block-based quantization achieves superior quantization accuracy. Furthermore, ZeRO++ convergence remains closely aligned with the baseline even when using 6-bit quantization.

**Fine-tuning.** We fine-tuned the pre-trained models: OPT-1.3B/-13B (Zhang et al., 2022b) and LLaMA-30B (Touvron et al., 2023), with ZeRO++ using FP6, INT4, and INT2 precision for $qwZ$ and INT-4 for $qgZ$ on the high-quality open-source instruction datasets[1]. We kept all training hyperparameters the same across all setups. The evaluation relies on the metric – perplexity (the lower, the better).

---

[1]We include the huggingface datasets: Dahoas/rm-static, Dahoas/full-hh-rlhf, Dahoas/synthetic-instruct-gptj-pairwise, yitingxie/rlhf-reward-datasets.

Table 3: End-to-end performance when using ZeRO++ w.\wo. optimized kernels.

|  | Optimized Quantization Kernel | Optimized Fusion Kernel | TFLOPs |
|---|---|---|---|
| Baseline | N/A | N/A | 15 |
| ZeRO++ | No | No | 19.73 |
| ZeRO++ | No | Yes | 21.6 |
| ZeRO++ | Yes | No | 31.40 |
| ZeRO++ | Yes | Yes | **36.16** |

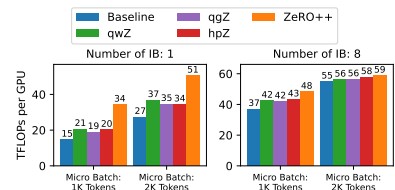

Figure 10: Throughput of 18B models on128 GPUs with ZeRO++, qwZ, qgZ, hpZ, and baseline on various InfiniBand connections.

Table 2 reveals the robustness of ZeRO++, as well as its ability to create low-precision models during fine-tuning without requiring post-quantization. Notice, the validation perplexity of ZeRO++ varies only slightly from the baseline, with a mere 0.27%-0.41% deviation, even when quantized to 4-bits.

### 4.4 THROUGHPUT BREAKDOWN AND ANALYSIS

**Impact of Individual Techniques.** In Figure 10, we show the individual and combined impact of qwZ, hpZ, and qgZ, on the throughput of 18B model on 128 GPUs. On low bandwidth clusters, each of these techniques enables a speedup ranging from 1.3-1.4x compared with baseline, while achieving an aggregated speedup of up to 2.26x. Note that our TFLOPs throughput is calculated from wall-clock time measurement, ZeRO++ aggregated throughput gain is not equivalent to sum of qgZ, qwZ, hpZ gain. We can validate the theoretical speedup with composition of our techniques by accumulating the speedup multiplicatively: $1.4 * 1.26 * 1.3 = 2.29$, which is very near to what we achieved as 2.26x.

For high bandwidth clusters, the individual speedup ranges between 1.13-1.16x, for a combined speedup of up to 1.3x. The figure demonstrates that each of these techniques has a similar impact towards throughput improvement and they compose effectively with a larger aggregated speedup.

**Impact of Kernel Optimizations.** We evaluate our optimized kernels on ZeRO++ throughput using an 18B model running on 64 GPUs.

Quantization Kernel: As shown in Table 3, compared with the baseline that uses PyTorch quantization, our optimized quantization kernels can achieve up to 1.67x speedup in terms of end-to-end throughput. Also, the baseline implementation suffers performance degradation as the group number increases which means the throughput gap will be larger when used with larger models.

Kernel Fusion: As described in Appendix C.2, kernel fusion is one of our key optimizations to improve memory throughput when executing sequences of CUDA kernels. Our fusion includes 1) tensor-reorder and quantization fusion 2) intra-node dequant, intra-node reduction and inter-node quant fusion. As shown in Table 3, we achieve up to 1.15x speedup on the end-to-end throughput.

## 5 CONCLUSION

This paper presents ZeRO++, an efficient collective communication solution for large model training using ZeRO stage-3. We optimize both model weights and gradients communication in the forward and backward pass of each training iteration. To reduce communication volume of model weights in forward propagation, we adopt block-based quantization and data pre-fetching. To remove cross-node communication of weights during backward pass, we hold secondary model partition on each node to trade memory for communication. To minimize gradient communication during backward propagation, we design and implement a novel all-to-all based gradient quantization and reduction scheme.

By incorporating all the three optimizations above, we improve system scalability and throughput for pre-training, fine-tuning and RLHF training with speedup of up to 3.3x. Furthermore, ZeRO++ can automatically quantize the parameters to ultra-low precision making the resulting model inference ready without post-training quantization. We envision ZeRO++ as the next generation of easy-to-use framework for training large models at scale.

ACKNOWLEDGMENTS

We would like to thank all DeepSpeed members for their helpful discussion and support to this work, and anonymous reviewers for their constructive feedback. This work was sponsored in part by NSF grant CAREER-2048044.

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

## APPENDIX

## A FURTHER BACKGROUND AND RELATED WORK

### A.1 DATA, TENSOR AND 3D PARALLELISM

Data parallelism (DP), pipeline parallelism (PP), and tensor parallelism (TP) are three forms of parallelism used to train large models across multi-GPU clusters. (Dean et al., 2012; Narayanan et al.,

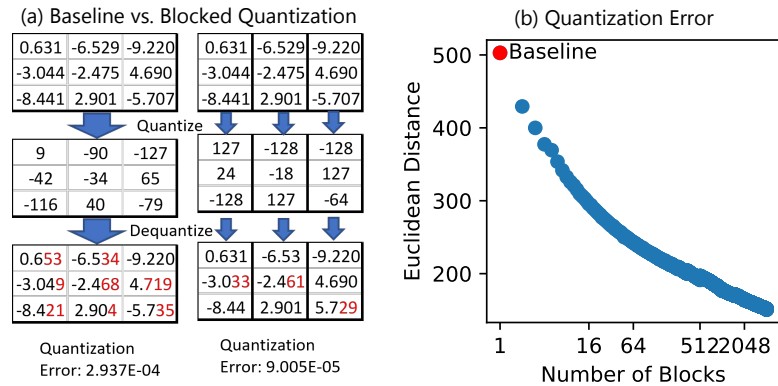

Figure 11: Illustration & example of block based quantization vs. baseline

2021; 2019; Huang et al., 2018) DP is commonly used when model size fits within a single GPU memory. In DP, each GPU holds a full copy of model weights and trains on separate input data. MP is orthogonal to DP, and is often used in cases where model size cannot fit into a single GPU's memory. Instead of splitting input data, model parallelism partitions a full model into pieces and assigns each model piece onto a GPU. There are mainly two approaches for model parallelism: i) pipeline parallelism (PP) and ii) tensor parallelism (TP). PP (Huang et al., 2019; Narayanan et al., 2019; Huang et al., 2018) splits models vertically, creating sequential stages consisting of a contiguous subset of layers. While there is sequential dependency between stages for an input micro-batch, the stages can be executed in parallel across micro-batches. In contrast, TP (Narayanan et al., 2021) splits each layer across multiple GPUs, where each GPU works on a different part of the layer for the same input.

3D parallelism (Smith et al., 2022; Team & Majumder, 2020) refers to combination of Data Parallelism , Pipeline Parallelism , and Tensor Parallelism (Dean et al., 2012; Narayanan et al., 2021; 2019; Huang et al., 2018), and is capable of achieving excellent throughput and scalability, and has been used to train a wide range of large language models (Microsoft, 2020; Narayanan et al., 2021; Radford et al., 2019; Black et al., 2022). Despite being highly efficient, 3D parallelism is severely limited by the fact that it requires complete rewrite of model and training pipeline to make them compatible with 3D parallelism (Smith et al., 2022).

## A.2 OTHER COMMUNICATION REDUCTION TECHNIQUES

**Quantization** In addition to the quantization approaches discussed in the main text, another line of research focus on error-feedback techniques to achieve efficient quantization without significant accuracy loss, such as (Tang et al., 2021; Wu et al., 2018). We view error-feedback as an orthogonal approach to ZeRO++.

**ZeRO Communication Reduction** Recent work such as QSDP (Markov et al., 2023) shows promising results using quantization of weights and gradients. Compared to these approaches, ZeRO++ uses a novel communication design to reduce quantization error, overlapping communication and quantization to hide quantization overhead, and a optimized weight placement to reduce inter-node communication, which is reflected in the non-trivial speedup in the evaluation.

## B DESIGN DETAILS

### B.1 BLOCK-SIZE BASED QUANTIZATION

As illustrated in Figure 11, each weight tensor is divided into smaller chunks, and converted into INT8 by symmetric quantization, using an independent quantization scaling coefficient. By keeping the quantization granularity small, we significantly mitigate the gap in number ranges and granularity.

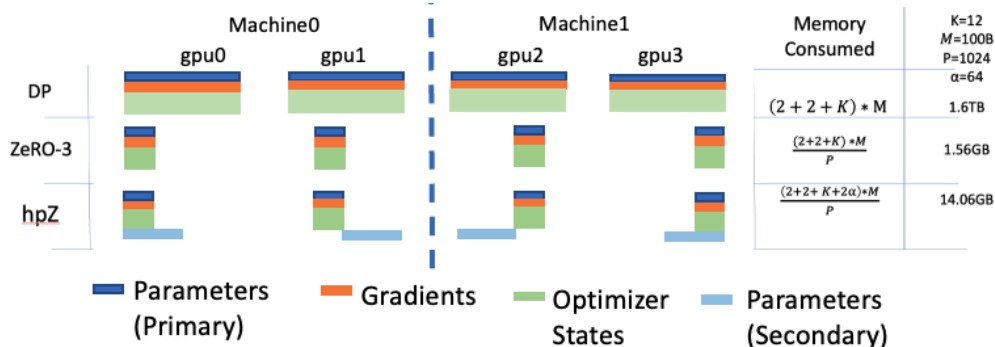

Figure 12: Per-device memory consumption analysis of standard data parallel (DP), ZeRO stage 3 (ZeRO-3) and proposed hierarchical partitioning of ZeRO parameters ($hpZ$). $K$ denotes the memory multiplier of optimizer states, $M$ represents the number of trainable parameters, $P$ is the data parallel group size or world size, and $\alpha$ is the number of secondary groups or ratio of world size to the number of ranks in the secondary group. A typical real world scenario example is provided in the last column. We assume a model size of 100B trained on 1024 V100 GPU DGX cluster (64 compute nodes, 16 GPUs per node).

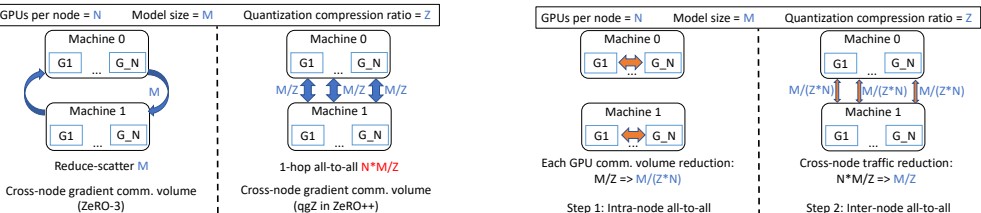

Figure 13: Compare the communication volume of ZeRO-3 reduce-scatter with the qgZ 1-hop all-to-all.

Figure 14: qgZ apply hierarchy all-to-all to reduce cross node traffic.

We show an example of the quantization error of performing block based quantization vs. the non-blocked quantization baseline in Figure 11(a). Figure 11(b) shows a case study of weights quantization on BERT model, where block based quantization reduces the quantization error by 3x. More in-depth convergence evaluations are shown in Section 4.

## B.2 MEMORY USAGE ANALYSIS OF $hpZ$

By design, $hpZ$ trades memory for communication efficiency. It is important to analyze this tradeoff. Recall that standard data parallel DNN (DP) replicates model parameters across data parallel ranks, ZeRO-3 on the other hand partitions parameter across data parallel ranks. A midway approach is model parameters partitioned across a subset of devices as long as model parameters fit.

Figure 12 provides a concrete memory usage estimate of a typical large language model of size of 100B parameters, with primary group size of 1024 GPUs and secondary group size of 16 GPUs (e.g., DGX-2 V100 node). As shown in Figure 12, with our proposed method, $hpZ$ consumes $8.9x$ more memory than ZeRO-3, our approach is still $114x$ less memory requirement than standard DP. This marginal increase in memory usage is compensated for by efficient intra-node communication schedule. By eliminating or reducing inter-node communication for backward pass, $hpZ$ reduces the end-to-end communication of ZeRO by $1.5x$, while supporting model training with hundreds of billions of parameters.

## B.3 REDUCING INTER-NODE COMMUNICATION VOLUME

Although replacing reduce-scatter with all-to-all achieves single-shot quantization and dequantization, it introduces a new problem; the inter-node communication volume increases instead of decreasing despite the quantization of data. We elaborate on this in Figure 13.

Given that intra-machine often have high bandwidth interconnects (e.g., NVLink, NVSwitch), cross-machine communication links are often the bottleneck. Given this, we analysis cross node communication volumes and ignore intra-node communication volumes.

Here we assume model size of $M$, GPU per node is $N$, gradient compression ratio as $Z$. Reduce-scatter, reduces the data during transmission over the ring, thus the total amount of data for cross-node communication is M. However, when using our 1-hop all-to-all approach, even though the data are compressed before communication (i.e., $M/Z$), each GPU needs to send out $M/Z$ amount of data to GPUs on the other nodes. Therefore, each machine will generate $N * M/Z$ amount of cross-node communication data, which is much bigger than reduce-scatter communication volume.

To address this, we do a hierarchical 2-hop all-to-all instead of 1-hop: a) first intra-node all-to-all and b) followed by inter-node all-to-all, which is shown as Figure 14. First, with high-bandwidth links among GPUs inside a machine, we conduct intra-node all-to-all on quantized data, then dequantize data and reduce on dequantized data. After intra-node quantization, all-to-all, dequantization, and reduction, we reduce the data size per GPU from $M/Z$ to $M/(Z * N)$. After intra-node all-to-all is completed, we conduct the inter-node all-to-all communication, which is similar to 1-hop all-to-all we described above. Given that now each GPU only needs to send out $M/(Z * N)$ data, the communication volume per machine is now $M/(Z * N) * N = M/Z$. By adopting this hierarchical all-to-all communication as 2-hop approach, we resolve the communication volume blow-up issue in our 1-hop scheme perfectly. Note that even though the total communication volume is doubled (one intra-node, the other inter-node), intra-node communication introduces negligible overhead given NVLink/NVswitch high bandwidth, and cross-node traffic has been significantly reduced, which is the major bottleneck in gradient communication.

## B.4 ZeRO++ Communication Volume Analysis

| Comm. Volume | forward all-gather | backward all-gather | backward reduce-scatter |
|---|---|---|---|
| ZeRO-3 | M | M | M |
| ZeRO++ | 0.5M | 0 | 0.25M |

Table 4: Communication volume comparison between ZeRO-3 and ZeRO++.

Table 4 illustrates theoretical communication volume comparison between ZeRO-3 and ZeRO++. We assume the model size of $M$. As described in Section 2, during ZeRO-3 there are 3 collective calls: all-gather on weights in forward pass, then all-gather on weights in backward pass and last is reduce-scatter on gradients in the backward. And each collective communicates $M$ volume of data.

With ZeRO-3, in total we need to communicate 3M data per each training iteration. Given that intra-node communication is fast with NVLink and NVSwitch, we ignore intra-node communication and focus on cross-node traffic only. For all-gather in the forward pass, by incorporating our quantized weights communication, we reduce communication volume from M to 0.5M. During the all-gather in the backward pass, by holding secondary weights partition within each node, we completely removed cross-node traffic. For reduce-scatter in the backward pass, by replacing reduce-scatter with our novel quantized gradient communication protocol, we reduce cross-node traffic from M to 0.25M. Therefore, compared with ZeRO-3, ZeRO++ reduces communication volume from 3M down to 0.75M for each training iteration.

## C Optimized Implementation

In this section, we discuss two key optimizations that enable ZeRO++ to fully realize the potential of 4x communication volume reduction to improve throughput without getting limited by implementation overheads: i) overlapping different communication and compute streams, when doing so enables better resource utilization, and ii) optimized CUDA kernels for quantization, dequantization, and tensor slice reordering operators, and kernel fusion across these operators when appropriate to minimize the memory traffic overhead. Below we discuss the two lines of optimization in detail.

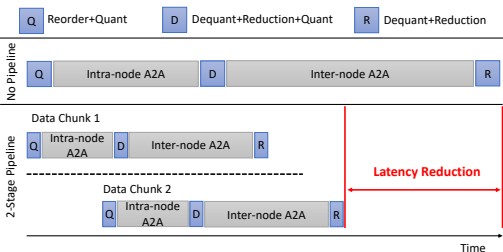

Figure 15: Pipelining and overlapping intra-node communication with inter-node communication in $qgZ$.

## C.1 OVERLAP COMPUTE AND COMMUNICATION

To reduce end-to-end communication time, we overlap quantization computation with communication for all-gathering of weights in both forward and backward passes. For the hierarchical all-to-all based reduce-scatter implementation of gradients, we overlap the intra-node communication with inter-node communication.

### C.1.1 COMMUNICATION-COMPUTATION OVERLAPPING ON WEIGHTS

For all-gather on weights, we enable communication-computation overlap using two key features : i) we track the execution order of model layers to get the sequence they will be fetched. ii) we guarantee asynchronous quantization execution. Specifically, the call to the quantization kernel is non-blocking and we further avoid operations that involve explicit/implicit CUDA synchronization (e.g. tensor concatenation), making the quantization a non-blocking operation that can be launched asynchronously.

With this two features, as ZeRO fetch parameters for each layer, the communication of the current layer and the quantization of the next layer can be launched at the same time on different CUDA streams. When the quantized data are needed for the next layer, ZeRO++ synchronizes the quantization stream to make sure the quantized data are ready. This approach hides the quantization cost of the next layer under the communication time span of the current layer which hides the quantization overhead.

### C.1.2 HIERARCHICAL COLLECTIVES FOR GRADIENT COMMUNICATION

As discussed in Section B.3, our all-to-all based gradient communication is broken into two stages: first intra-node communication followed by inter-node communication. The inter-node communication depends on the results of the intra-node communication, therefore, with a naive implementation, inter-nodes links are idle during intra-node communication and vice versa. To reduce latency by leveraging both inter-node and intra-node links in parallel, we chunk our input gradient tensor and pipeline transfer between intra-node communication and inter-node communication. As shown in Figure 15, compared with "no pipeline" case on the top, simply adopting a "2-stage pipeline" transfer achieves the amount of end-to-end latency reduction shown as the red arrow-line in Figure 15. By overlapping intra-node and inter-node communication, the end-to-end latency of gradient communication is significantly reduced.

Doing this pipeline correctly has implications on our tensor slice reordering process. The more pipeline stages we have, the more fine-grained tensor slices are needed for reordering. Therefore, we also propose a generalized tensor slices reordering scheme as algorithm 2, which covers both w/ and w/o pipelining data transfer cases. Here stages refer to the number of pipeline stages we have, nodeSize is the number of GPUs per node and nodes is the number of nodes.

Next, we discuss how we optimize our CUDA kernels to further reduce all quantization related overhead.

---

**Algorithm 2:** Generalized tensor slice reordering ($qgZ$)

---

**Constants:** $stages, nodeSize, nodes$
**Input**     : $partitionID$
**Output**    : $mappedPartitionID$

1  $totalDevices \leftarrow nodeSize * nodes$;
2  $stageID \leftarrow partitionID \% stages$;
3  $chunkID \leftarrow \frac{partitionID}{stages}$;
4  $pipelineOffset \leftarrow stageID * totalDevices$;
5  $chunkOffset \leftarrow \frac{chunkID}{nodeSize}$;
6  $chunkBase \leftarrow (chunkID \% nodeSize) * nodes$;
7  **Return:** $pipelineOffset + chunkBase + chunkOffset$;

---

## C.2 CUDA KERNELS

As existing quantization implementations are unable to capture the combination of data mapping and high throughput necessary to minimize kernel overhead, we implement and optimize custom CUDA kernels to implement these primitives. In particular, these kernels aim to (1) saturate device memory bandwidth and (2) minimize the total traffic via fusion.

**Maximizing Bandwidth Utilization:** A core quantization and dequantization library of composable operators was developed as the foundation for ZeRO++. The core primitives leverage efficient vectorized memory accesses at the maximum granularity a given GPU architecture supports. In order to satisfy the alignment requirements these instructions have, model state is partitioned such that quantization granularities will be 16B aligned. Additionally, we leverage instruction level parallelism to overlap multiple memory transactions with each other. In practice, the combination of vectorized accesses and instruction level parallelism enables the quantization library to achieve full GPU memory bandwidth utilization.

**Minimizing Total Traffic:** Multiple techniques are used to reduce the total memory traffic for quantization kernels. First, the size of each quantization block is tuned so as to express sufficient parallelism to schedule across a GPU's streaming multiprocessors and cache values not quantized yet in the register file while calculating the quantization scale and offset for the block. Second, we fuse tensor reshaping and quantization into the same kernel to avoid redundantly loading data from global memory. For example, the tensor slice reordering (i.e., orange arrow-lines in Figure 5) is realized within a fused quantization and remapping kernel.This fused kernel achieves the same level of performance as a single quantization kernel working with contiguous data. Finally, we fuse sequential dequantization, reduction, and quantization operations into single kernel implementation, which reduces total memory traffic by 9x in $qgZ$.

# D ADDITIONAL EVALUATION & DISCUSSION

## D.1 EXTENDED CONVERGENCE DISCUSSION

In this section, we present the raw data of our convergence evaluation for both pretraining and fine-tuning in Table 5 and 6. Please note the quantization bits shown here refer to the weight quantization. For the gradient, we use INT4 for all the experiments.

For theoretical convergence analysis, one thing to note is the unique design of ZeRO. In ZeRO paradigm, there are two kinds of weights: the temporary weights, which are all-gathered from all GPUs in order to do forward/backward on a particular model layer and then discarded; and the local weight shard, which is used to populate the temporary weights, are permanent and will be updated at each optimizer step. The ZeRO++ quantization only happens when we communicate the temporary weights. Thus, the local shard of the weights will be free of the quantization error. For the gradients that are computed from the quantized temporary weights, there will be minor errors for 8-bits and even 4 bits.

Table 5: Convergence in terms of validation lm loss when pre-training with different quantization bits.

| Model: GPT 350M | | |
|---|---|---|
| Iteration | FP16 | INT8 |
| 2400 | 2.891633 | 2.89641 |
| 4800 | 2.595669 | 2.600295 |
| 7100 | 2.50512 | 2.509532 |
| 9500 | 2.409789 | 2.417458 |
| 11800 | 2.355472 | 2.366978 |
| 14100 | 2.331281 | 2.345854 |
| 16500 | 2.286052 | 2.303065 |
| 18800 | 2.279778 | 2.301165 |
| 21200 | 2.244547 | 2.26685 |
| 23400 | 2.220746 | 2.246421 |

| Model: GPT 13B | | | |
|---|---|---|---|
| Iteration | FP16 | INT8 | FP6 |
| 2400 | 3.346583 | 3.340124 | 3.327485 |
| 4800 | 2.794749 | 2.787855 | 2.789709 |
| 7100 | 2.56617 | 2.561561 | 2.562001 |
| 9500 | 2.398518 | 2.399067 | 2.39818 |
| 11800 | 2.370753 | 2.358937 | 2.357589 |
| 14100 | 2.310235 | 2.33077 | 2.329953 |
| 16500 | 2.275294 | 2.277154 | 2.275939 |
| 18800 | 2.180487 | 2.239978 | 2.238353 |
| 21200 | 2.181295 | 2.194993 | 2.193342 |
| 23400 | 2.181079 | 2.184269 | 2.183024 |

Table 6: Convergence in terms of evaluation perplexity when fine-tuning with different quantization bits.

| Model: OPT-1.3b | | | | |
|---|---|---|---|---|
| Epoch | FP16 | FP6 | INT4-bit | INT2 |
| 1 | 2.018 | 2.021 | 2.034 | 2.430 |
| 2 | 1.972 | 1.975 | 1.989 | 2.355 |
| 3 | 1.934 | 1.938 | 1.953 | 2.320 |
| 4 | 1.906 | 1.910 | 1.924 | 2.291 |
| 5 | 1.882 | 1.888 | 1.905 | 2.273 |
| 6 | 1.863 | 1.868 | 1.887 | 2.267 |
| 7 | 1.850 | 1.856 | 1.871 | 2.248 |
| 8 | 1.842 | 1.841 | 1.861 | 2.241 |
| 9 | 1.832 | 1.830 | 1.850 | 2.228 |
| 10 | 1.822 | 1.833 | 1.844 | 2.226 |
| 11 | 1.818 | 1.821 | 1.838 | 2.224 |
| 12 | 1.809 | 1.817 | 1.833 | 2.223 |
| 13 | 1.805 | 1.811 | 1.832 | 2.218 |
| 14 | 1.804 | 1.809 | 1.835 | 2.221 |
| 15 | 1.807 | 1.809 | 1.832 | 2.222 |
| 16 | 1.809 | 1.812 | 1.829 | 2.216 |

| Model: OPT-13b | | | | |
|---|---|---|---|---|
| Epoch | FP16 | FP6 | INT4-bit | INT2 |
| 1 | 1.813 | 1.821 | 1.821 | 1.842 |
| 2 | 1.698 | 1.705 | 1.705 | 1.705 |
| 3 | 1.724 | 1.717 | 1.736 | 1.734 |

| Model: LLaMA-30b | | | | |
|---|---|---|---|---|
| Epoch | FP16 | FP6 | INT4-bit | INT2 |
| 1 | 1.57 | 1.58 | 1.580 | 2.524 |
| 2 | 1.49 | 1.50 | 1.494 | 1.664 |
| 3 | 1.64 | 1.63 | 1.601 | 1.544 |

| Model Size | Token Size | ZeRO TFLOPs | hpZ TFLOPs | MiCS TFLOPs |
|---|---|---|---|---|
| 7.5B | 1K | 36.99 | 38.39 | 38.96 |
| 7.5B | 2K | 53.3 | 54.4 | 52.72 |
| 18B | 1K | 51.47 | 52.42 | OOM |
| 18B | 2K | 60.94 | 61.44 | OOM |

Table 7: hpZ vs MiCS evaluation on a 4 node cluster (16 V100 GPUs per node)

| 64 V100 GPUs Ethernet 10Gbps | | | | |
|---|---|---|---|---|
| Model Size | Tokens per GPU | Baseline TFLOPs | ZeRO++ TFLOPs | Speedup |
| 18B | 2K | 9.72 | 31.94 | 229% |
| 18B | 1K | 4.98 | 19.16 | 284% |
| 18B | 512 | 2.51 | 10.03 | 299% |
| 7.5B | 2K | 10.07 | 24.51 | 143% |
| 7.5B | 1K | 4.81 | 15.90 | 231% |
| 7.5B | 512 | 2.48 | 7.32 | 195% |

Table 8: Evaluation of ZeRO++ pretraining throughput on Ethernet with 10Gbps bandwidth

In terms of the theoretical convergence with gradient noise/error, existing research such as (Gorbunov et al., 2020; Ramezani-Kebrya et al., 2021; Alistarh et al., 2017) prove minor disturbance on gradients will not impact the overall convergence (the convergence rate follows O(1/sqrtT) for T iterations), which we have empirically show.

We also see error-feedback as a kind of orthogonal approach with ZeRO++ and we believe adding error-feedback technique such as (Tang et al., 2021; Wu et al., 2018) could help especially for 1-bit/2-bit training or for a smaller model, which is a part of our future work.

## D.2 COMPARING HPZ WITH MICS

As previously discussed in Appendix 2, closely related to hierarchical weight partition for ZeRO ($hpZ$) is $MiCS$(Zhang et al., 2022c). The key difference between the two methods is what data are replicated in the secondary group; only model weights are replicated in $hpZ$, while entire model states are replicated in $MiCS$. Table 7 shows the per-GPU throughput of both methods for different model and token size configurations. The table also shows that given a secondary partition size of a single node (16 V100 GPUs), $hpZ$ can support 18 billion parameter model whereas $MiCS$ reports out-of-memory (OOM) at this scale.

## D.3 ETHERNET CLUSTER EVALUATION

In the main text, we present low-bandwidth evaluation with 100Gbps bandwidth. Here we further extend our evaluation to 10Gbps Ethernet scenario to show how inter-node bandwidth affects ZeRO++ performance in a more comprehensive manner. The pretraining setting is the same, where weights are quantized into INT8 and gradients are quantized into INT4. As Table 8 shows, ZeRO++ can outperform baseline by up to 4x in terms of throughput on 10Gbps Ethernet, and the performance pattern is consistent with previous evaluation with InfiniBand. This aligns with our evaluation analysis in the main text that the performance boost of ZeRO++ increases as network bandwidth decreases, making ZeRO++ a valuable tool for training in low-bandwidth clusters.

