# OpenReview forum: "ZeRO++: Extremely Efficient Collective Communication for Large Model Training"
_ICLR.cc/2024/Conference — ICLR 2024 poster_

### Official Review · Reviewer_QbwT · 2023-10-31

**Soundness:** 4 excellent
**Presentation:** 3 good
**Contribution:** 3 good
**Rating:** 8
**Confidence:** 3

**Summary:**

This paper proposes zero++, which contains 3 communication reduction techniques (low-precision parameter all-gather, data remapping, and low-precision gradient averaging), in order to reduce the communication overhead and acceleration the training.

**Strengths:**

1. This paper proposes zero++, which contains 3 communication reduction techniques (low-precision parameter all-gather, data remapping, and low-precision gradient averaging) with sufficient details provided in the appendix.

2. Additional to the system design, an optimized implementation is also proposed.

3. The experiments show good performance on real-world training tasks.

**Weaknesses:**

1. Some part of the paper is a little bit vague and confusing, please refer to the "Questions" below for more details.

2. The main reason I'm holding back from a better score is that zero++ seems to have huge overlap with QSDP (Markov, I., Vladu, A., Guo, Q., & Alistarh, D. (2023). Quantized Distributed Training of Large Models with Convergence Guarantees. ICML 2023). I understand that QSDP was officially published on ICML not very long ago and we reviewers should not take the release date on arxiv as a reason to reject this paper. However, this is a delicate situation for me and I guess we will need the AC to judge.
For now, I still need the authors to give a detailed comparison between QSDP and zero++, which will help a lot for my paper review.

**Questions:**

1. Does the gradient and parameter communication compression use the same quantizer/quantization algorithms? What are the specific quantization algorithm used in the experiments?

2. In experiments, some details are missing. For example, in Section 4.3 "Pretraining" (or Figure 9), it is mentioned that 8-bit or 6-bit quantization is used. However, it is unknown whether is 8-bit for both parameters and gradients? or 8-bit for parameters and fp16 for gradients? or fp16 for parameters and 8-bit for gradients? I strongly recommend the authors to specify what (parameter or gradient) is quantized wherever quantization/precision configuration is mentioned.

3. In the experiments,  Section 4.3 "Fine-tuning" (or Table 2), it is mentioned that "with ZeRO++ using FP6, INT4, and INT2 precision for qwZ and INT-8 for qgZ ..." I wonder why not also conduct some experiments of lower precision (int4, int2) on qgZ? Also, is there any ablation experiments on other configurations, such as int4 qwZ + fp16 qgZ?

BTW, there is "FP6" in Section 4.3 (see the sentence I quoted above), is it a typo and the authors actually mean "FP16" here?

4. What's the difference between Zero++ and QSDP (in system design), except for the hierarchical partitioning in Zero++ (hpZ)?

---

> ### Author Response · Authors · 2023-11-17
>
> Thank you for your insightful feedback. We have carefully considered your comments and addressed them in the following response.
>
> 1.	The main reason I'm holding back from a better score is that zero++ seems to have huge overlap with QSDP (Markov, I., Vladu, A., Guo, Q., & Alistarh, D. (2023). Quantized Distributed Training of Large Models with Convergence Guarantees. ICML 2023). I understand that QSDP was officially published on ICML not very long ago and we reviewers should not take the release date on arxiv as a reason to reject this paper. However, this is a delicate situation for me and I guess we will need the AC to judge. For now, I still need the authors to give a detailed comparison between QSDP and zero++, which will help a lot for my paper review. What's the difference between Zero++ and QSDP (in system design), except for the hierarchical partitioning in Zero++ (hpZ)?
>
> A: The common part of QSDP and ZeRO++ lies in both using block-based quantization for weights. However, there are three major differences between these two approaches.
> * i) concurrency. Although using block-based quantization will help mitigate quantization error. The cost of doing it will cancel out most of the benefit from communication reduction in terms of wall-clock time. ZeRO++ hide quantization wall-clock time by overlapping quantization with subsequent communication. More specifically, ZeRO++ put weights into chunks and then overlapping the communication of current chunk with quantization of next chunk to hide the quantization cost (as well as more optimizations discussed in Appendix C). This is directly reflected in the evaluation: for 100Gbps bandwidth, ZeRO++ can achieve 130% speedup over the baseline whereas QSDP achieves 15% speedup. We also extended our evaluation to 10Gbps scenarios where ZeRO++ achieves 300% speedup.
> * ii) gradients. We observe that performing reduce-scatter on quantized gradients, as QSDP does, will not average out the quantization error but amplify them, resulting in non-trivial convergence gap. Thus ZeRO++ proposes a novel all2all primitive to make sure communicating quantized gradients will not amplify quantization error. The convergence eval shows the gap, for 1.3B model, QSDP shows 0.34 gap in perplexity when both weights and gradients are quantized to 8 bits while ZeRO++ shows 0.005 gap when the weights are quantized to 6 bits and gradients are quantized to 4 bits. On the performance side, we also overlapped intranode all2all with internode all2all with proper synchronization to further reduce end-2-end latency, which is detailed in appendix C.1.2
> * iii) optimized placement of weights. As reviewer mentioned, ZeRO++ equips with hpZ to completely get rid of all inter-node communication in backward to further boost system performance.
>
> We will extend our related work part to add the discussions to QSDP.
>
> 2.	Does the gradient and parameter communication compression use the same quantizer/quantization algorithms? What are the specific quantization algorithm used in the experiments?
>
> A: We use block-based quantization for both weights and gradients. The details of it can be found in Appendix B.2. Simply put, we pre-scale the tensors by small chunks before casting them to lower precision to mitigate the difference in numerical ranges and granularities.
>
> 3.	In experiments, some details are missing. For example, in Section 4.3 "Pretraining" (or Figure 9), it is mentioned that 8-bit or 6-bit quantization is used. However, it is unknown whether is 8-bit for both parameters and gradients? or 8-bit for parameters and fp16 for gradients? or fp16 for parameters and 8-bit for gradients? I strongly recommend the authors to specify what (parameter or gradient) is quantized wherever quantization/precision configuration is mentioned.
>
> A: For the experiments in the paper, we use INT4 as the quantization format for gradients. The weights are quantized into INT8 in most testing scenarios. We also include 6, 4, 2 bits for other scenarios where we state in text to showcase how even lower-bit affects the convergence. For all the evaluation, we enable all three optimization techniques (qwZ, qgZ, hpZ) so both the gradients and the weights are quantized except for the ablation study. We will update the text in our camera-ready version to clarify this.
>
> (To be continued)

---

> > ### Author Response · Authors · 2023-11-17
> >
> > 4.	In the experiments, Section 4.3 "Fine-tuning" (or Table 2), it is mentioned that "with ZeRO++ using FP6, INT4, and INT2 precision for qwZ and INT-8 for qgZ ..." I wonder why not also conduct some experiments of lower precision (int4, int2) on qgZ? Also, is there any ablation experiments on other configurations, such as int4 qwZ + fp16 qgZ?
> >
> > A: For all the experiments in the paper, we use INT4 as the quantization format for gradients. The “INT-8 for qgZ” here is a textual error. We will correct this error. We do plan to try even more extreme formats like INT2 for gradients in our future work.
> >
> > 5.	BTW, there is "FP6" in Section 4.3 (see the sentence I quoted above), is it a typo and the authors actually mean "FP16" here?
> >
> > A: "FP6" is accurate. We use block-based quantization to pre-scale the tensors and then cast it to FP6 with E3M2 format. Our purpose is to highlight our framework works for both floating-point quantization and integer quantization.
> >
> > We welcome any further questions or points of discussion.

---

### Official Review · Reviewer_ZpeK · 2023-11-01

**Soundness:** 3 good
**Presentation:** 3 good
**Contribution:** 2 fair
**Rating:** 5
**Confidence:** 5

**Summary:**

This paper proposed ZeRO++, which includes three main techniques to accelerate communication in distributed training of foundation models: i) low-precision AllGather for parameter collection, ii) hierarchy communication optimization, and ii) low-precision gradient aggregation. Empirical studies was conducted to verify the performance of the proposed method.

**Strengths:**

- Optimizing the communication in ZeRO or more generally speaking, optimizer parallelism is an important problem to scale out / speed up foundation model training.

- The proposed design and implementation are reasonable.

- The evaluation is conducted on a large production-level GPU cluster to evaluate models at the state-of-the-art scales.

**Weaknesses:**

- My main concern about the paper is lack of novelty as a research paper instead of a software report, concretely, w.r.t each technique contribution, I have the following comments:
  - Low-precision gradient averaging: such a technique has been studied for decades in standard data parallel training; integrating them in an existing system probably cannot be viewed as a novelty in a top ML volume (i.e., ICLR).
  - Organizing the communication as a hierarchy topology is not a new approach; a similar implementation was introduced in PyTorch-FSDP about six months ago.
  - Distributing the model weights with lower precision is an interesting point. However, there is a lack of concrete theoretical analysis about this approach, i.e., now the model parameter includes some quantization error; how would this influence the convergence behavior?

**Questions:**

Would it be possible to provide some theoretical analysis about the convergence given the quantized communication of the weights distribution?

---

> ### Author Response · Authors · 2023-11-17
>
> Thank you for your insightful feedback. We have carefully considered your comments and addressed them in the following response.
>
> 1.	Low-precision gradient averaging: such a technique has been studied for decades in standard data parallel training; integrating them in an existing system probably cannot be viewed as a novelty in a top ML volume (i.e., ICLR).
>
> A: The novelty of ZeRO++ is not to use low-precision gradient but lies on using low-precision gradient without impacting the convergence and system throughput. Because even upon using the state-of-the-art quantization to mitigate the quantization error, it will still cause a significant gap in terms of convergence as well as throughput (due to the time spent on quantization). That is why we present a novel all-to-all communication primitive, and a full set of CUDA kernels, and a carefully designed system to i) maintain numeric precision even during communication in a quantized format; ii) minimize the quantization cost and iii) boost system throughput by doing intra-node and inter-node communication concurrently and hide quantization cost by doing communication and quantization concurrently.
>
> 2.	Organizing the communication as a hierarchy topology is not a new approach; a similar implementation was introduced in PyTorch-FSDP about six months ago.
>
> A: Our approach, hpZ, is similar to FSDP/HSDP in that they both trade off memory for communication. However, hpZ is different in that it creates hierarchical partitioning only for the parameters instead of all model states, resulting in much lower memory overhead. This allows hpZ (and ZeRO++) to support larger models than FSDP can. Appendix D of our manuscript shows our comparison with MiCS, which has the same design idea as FSDP. Our evaluation comparing MiCS with hpZ shows that hpZ can support an 18 billion parameter model, whereas MiCS reports OOM at this scale (see Table 7 of our manuscript).
>
> 3.	Distributing the model weights with lower precision is an interesting point. Distributing the model weights with lower precision is an interesting point. However, there is a lack of concrete theoretical analysis about this approach, i.e., now the model parameter includes some quantization error; how would this influence the convergence behavior? Would it be possible to provide some theoretical analysis about the convergence given the quantized communication of the weights distribution?
>
> A: Regarding theoretical convergence, one notable aspect is the unique design of ZeRO. In the ZeRO paradigm, there are two kinds of weights: the temporary weights, which are all-gathered from all GPUs in order to do forward/backward on a particular model layer and then discarded; and the local weight shard, which is used to populate the temporary weights, are permanent and will be updated at each optimizer step. The ZeRO++ quantization only happens when **we communicate the temporary weights**. Thus, the local shard of the weights will be free of the quantization error. For the gradients that are computed from the quantized temporary weights, there will be minor errors, yet fruitful research such as [1, 2, 3] prove that minor disturbance to gradients will not impact the overall convergence (the convergence rate follows O(1/sqrt{T}) for T iterations), which we have also empirically shown. An empirical evaluation can be found in Table 2 where the convergence gap increases as the quantization goes from 6 bits to 2 bits. We also added the above discussion to the appendix to better discuss ZeRO++’s theoretical convergence. Thank you for the constructive suggestions.
> >[1] Gorbunov, Eduard, Filip Hanzely, and Peter Richtárik. "A unified theory of SGD: Variance reduction, sampling, quantization and coordinate descent." AISTATS 2020
> >
> >[2] Ramezani-Kebrya, Ali, et al. "NUQSGD: Provably communication-efficient data-parallel SGD via nonuniform quantization." The Journal of Machine Learning Research 22.1 (2021): 5074-5116.
> >
> >[3] Alistarh, Dan, et al. "QSGD: Communication-efficient SGD via gradient quantization and encoding." Advances in neural information processing systems 30 (2017).
>
> We welcome any further questions or points of discussion.

---

> > ### Comment · Reviewer_ZpeK · 2023-12-04
> > **Thank you for your feedback!**
> >
> > Thank you for the additional information. However, I think the rebuttal does not fully address my concerns. I would keep my score.

---

### Official Review · Reviewer_V3tv · 2023-11-02

**Soundness:** 3 good
**Presentation:** 3 good
**Contribution:** 2 fair
**Rating:** 5
**Confidence:** 4

**Summary:**

This work, ZeRO++, proposes a collective communication optimization for ZeRO3 of DeepSpeed library. Three elements are introduced: 1) quantized AllGather of weight, 2) hierarchical partition, and 3) quantized ReduceScatter of gradient with AlltoAll. When combined together, ZeRO++ reduces communication volume and network hops to mitigate the communication bottleneck of ZeRO3 training for large models.

**Strengths:**

+. Replaced Ring-ReduceScatter with AlltoAll for lower latency and less quantization/dequantization times

+. Optimized intra-node and inter-node AlltoAll

**Weaknesses:**

-. **Limited contribution**
> qwZ quantizes FP16 weights to lower precision right during the all-gather, and dequantizes them back to FP16 on the receiver side, and then conducts layer computation.

* Why cast FP16 weight into INT4 is a novel contribution? Just use "torch.Tensor.to()"?
* What if model is being trained FP8 on the modern H100? so the communication benefit will be halved?
* Will cast FP16 into INT4 occupy 25%+ extra GPU memory (assuming zero fragmentation)?

> hpZ removes cross node traffic in backward all-gather by holding secondary weight partitions in on device memory.

* How is this scheme different from HSDP of PyTorch (Intra-Machine FSDP to shard weight + inter-Machine DDP to replicate weight)? HSDP was published in VLDB'2023.

> qgZ is a novel quantized reduce-scatter algorithm based on all-to-all collectives that enables a 4xcommunication volume reduction of gradient reduce-scatter by replacing FP16 with INT4.

* Same question as weight quantization

-. **Limited Convergence Analysis**

> Pretrain: We analyze the pretraining of GPT-350M and GPT-13B models on the Pile dataset

How does this work apply to different datasets for convergence behavior, considering extreme quantization?

Besides Figure 9, does this work provide accurate numbers for loss diffs for pertaining?

> Fine-tuning. Table 2

It seems this work always worse the perplexity compared with baseline.

**Questions:**

*. Does this work provide open-source code?

*. Is this work ever used in production for industrial models?

---

> ### Author Response · Authors · 2023-11-17
>
> Thank you for your constructive feedback. We have carefully considered your comments and addressed them in the following response.
>
> 1.	Why cast FP16 weight/gradient into INT4 is a novel contribution? Just use "torch.Tensor.to()"?
>
> A: Simply casting floating-point weights/gradients to integers will cause the training to crash from the very beginning due to the significant difference in numeric granularity (e.g. the usual granularity for fp16 is $10^{-6}$ vs. $1$ in integer format). Even upon using the state-of-the-art quantization approach to mitigate the gap, it will still cause a significant gap in terms of convergence as well as throughput (due to the computation time spent on quantization and dequantization). That is why we present a novel all-to-all communication primitive, and a full set of CUDA kernels, and a carefully designed system to
> * i) maintain numeric precision even during communication in a quantized format;
> * ii) minimize the quantization cost by custom CUDA kernels;
> * iii) boost system throughput by doing intra-node and inter-node communication concurrently and hide quantization cost by doing communication and quantization concurrently.
>
> 2.	What if model is being trained FP8 on the modern H100? so the communication benefit will be halved?
>
> A: ZeRO++ is proposed to tackle the challenge that communication has taken a gradually larger portion of running time as scalability increases. FP8 training will reduce both the computation time and communication time thus the challenge of communication ratio remains, which makes ZeRO++ a valid solution. By hpZ, we are eliminating all backward weight intra-node communication so its reduction will not be impacted by the weight format. For qgZ and qwZ, its reduction depends on the quantization format users prefer, which is why we show an extensive evaluation with different quantization formats.
>
> 3.	Will cast FP16 into INT4 occupy 25%+ extra GPU memory (assuming zero fragmentation)?
>
> A: The system implementation of ZeRO++ makes sure quantization&communication happens by chunks. That means the memory overhead is at most two chunks (one for comm and one for quantization) which is trivial compared with the overall model size. The chunk size is adjustable so the user can even squeeze the memory overhead to almost nothing (though we believe there is hardly such need).
>
> 4.	How is this hpZ scheme different from HSDP of PyTorch (Intra-Machine FSDP to shard weight + inter-Machine DDP to replicate weight)? HSDP was published in VLDB'2023.
>
> A: Our approach, hpZ, is similar to HSDP in that they both trade off memory for communication. However, hpZ is different in that it creates hierarchical partitioning only for the parameters instead of all model states, resulting in lower memory overhead. This allows hpZ (and ZeRO++) to support larger models than HSDP can. HSDP, by design, supports relatively medium-sized models, as noted by the authors. Appendix D of our manuscript shows our comparison with MiCS (Zhang et.al, MiCS: Near-linear Scaling for Training Gigantic Model on Public Cloud), which has the same design idea as HSDP. Our evaluation comparing MiCS with hpZ shows that hpZ can support an 18 billion parameter model, whereas MiCS reports out-of-memory (OOM) at this scale (see Table 7 of our manuscript).
>
> 5.	How does this work apply to different datasets for convergence behavior, considering extreme quantization?
>
> A: In our evaluation, we show a wide range of datasets including Pile, Dahoas/rm-static, Dahoas/full-hh-rlhf, Dahoas/synthetic-instruct, gptj-pairwise, yitingxie/rlhf-reward-datasets. We believe these extensive convergence results are enough to demonstrate the good convergence for ZeRO++ for general applications.
>
> 6.	Besides Figure 9, does this work provide accurate numbers for loss diffs for pertaining  ?
>
> A:  We have updated our manuscript to include Table 5 and 6, providing the precise loss differences to complement Figure 9 and Table 2 for both pretraining and finetuning. This addition will ensure our results are both transparent and comprehensive. Thank you for the suggestion.
>
> (To be continued)

---

> > ### Author Response · Authors · 2023-11-17
> >
> > 7.	From fine-tuning Table 2, it seems this work always worse the perplexity compared with baseline.
> >
> > A: The gap in perplexity arises because we use extreme quantization which inevitably introduces quantization error. What we do show is how marginal these errors are: a gap of 0.005 in terms of perplexity is impressive considering the model is quantized from fp16 to fp6 (with the gradients even compressed to int4). In comparison, related works such as QSDP shows perplexity gap of 0.34 even when doing less aggressive quantization (8bits for both weights and gradients).
> >
> > 8.	Does this work provide open-source code? Is this work ever used in production for industrial models?
> >
> > A: ZeRO++ has been released to the open-source community and it has been used in several other research works as well as industrial uses. Unfortunately we cannot share the link due to double-blind policy. In our paper, we show ZeRO++ working up to 30B models, which demonstrate its effectiveness for industrial models.
> >
> >
> > We welcome any further questions or points of discussion.

---

> > > ### Comment · Reviewer_V3tv · 2023-11-23
> > >
> > > Thank you for the rebuttal and clarification. The rebuttal has addressed a part of my concerns. I raised my score.

---

### Official Review · Reviewer_TkZG · 2023-11-11

**Soundness:** 3 good
**Presentation:** 3 good
**Contribution:** 3 good
**Rating:** 6
**Confidence:** 4

**Summary:**

The paper proposes an extension to the existing memory optimal training technique for LLMs, *ZeRO*. The proposed enhancements address the expensive communication problems of the state-of-the-art method, especially when trained on low bandwidth networks. ZeRO++ with the new quantization enabled communication strategies achieves reasonably good performance over ZeRO while keeping the performance of LLMs intact.

**Strengths:**

- The paper is well written, detailed and easy to understand for both DL and HPC communities. (assuming readers have some basic understanding of both the worlds).
- Has a good coverage of the existing literature and positions the paper with-in the body of work.
- The problem is probably impactful for low-resource devices such as low band-width networks.

**Weaknesses:**

- The paper shows the effectiveness of the proposed method on powerful DGX GPUs with infiniBand interconnects. Noway, we can consider this interconnect as low-bandwidth.
- Intra-node weight storage sounds interesting and empirically showed to work for upto 30B model fine-tuning and 13B model pre-training. Wonder, if this can be scaled if there is any ablation on this dimension?

**Questions:**

- One simple experiment (even on a network of RasberryPi machines) with low-badwidth networks will obviously greatly strengthens the paper claims/contributions. This can be fine-tuning does not even have to be pre-training.
- A feasibility study on the scalability of the method, especially because of the GPU memory tradeoff within the node, would help.
- Upon some or all of the above concerns being addressed, would be happy to raise the score.
- Also, wonder, if there is any open implementations for this complex custom communication MPI primitives?

## Post Rebuttal

- Raising the score to above the acceptance threshold

---

> ### Author Response · Authors · 2023-11-17
>
> Thank you for your insightful feedback. We have carefully considered your comments and addressed them in the following response.
>
> 1.	The paper shows the effectiveness of the proposed method on powerful DGX GPUs with infiniBand interconnects. Noway, we can consider this interconnect as low-bandwidth. One simple experiment (even on a network of RasberryPi machines) with low-badwidth networks will obviously greatly strengthens the paper claims/contributions. This can be fine-tuning does not even have to be pre-training.
>
> A: We define "low-bandwidth" in our context as 100Gbps to simulate the speed of Ethernet connections in typical cloud services and HPC clusters. Yet we agree with the reviewer that an evaluation for an even lower bandwidth cluster is needed. Thus, we extended our evaluation to an entry-level cluster with 10Gbps Ethernet connections and demonstrated that ZeRO++ can boost training throughput by 4x in such scenarios (see Appendix D for full results). This aligns with our previous evaluation, showing that ZeRO++'s speedup increases significantly as the inter-node bandwidth decreases.
>
> 2.	Intra-node weight storage sounds interesting and empirically showed to work for up to 30B model fine-tuning and 13B model pre-training. Wonder, if this can be scaled if there is any ablation on this dimension? A feasibility study on the scalability of the method, especially because of the GPU memory tradeoff within the node, would help.
>
> A: An analysis of the scalability of hpZ can be found in Appendix B.2 of our paper, with an example of a 100B model running on 1024 GPUs. In that section, we also provide mathematical models to calculate the memory usage for the given model size and number of GPUs.
>
> 3.	Also, wonder, if there is any open implementations for this complex custom communication MPI primitives?
>
> A: ZeRO++ has been made publicly available to the open-source community and has seen adoption in both research and industrial applications. Unfortunately, we are unable to share the link due to double-blind policy.
>
> We welcome any further questions or points of discussion.

---

### Official Review · Reviewer_kJiX · 2023-11-13

**Soundness:** 3 good
**Presentation:** 4 excellent
**Contribution:** 3 good
**Rating:** 6
**Confidence:** 4

**Summary:**

This paper proposes ZeRO++, a communication-efficient extension of the commonly used memory-efficient training framework for LLMs, ZeRO. The proposed techniques include three specific techniques to reduce communication costs, improve the throughput and accelerate the training process.

**Strengths:**

1. This paper proposes quantized weight communication, hierarchical partitioning, and quantized gradient communication techniques to reduce the communication costs and achieves efficient implementation of overlap computing and communication. The parallelism of training is the focus.

2. The experiments presented are quite convincing, including large-scale language model training and fine-tuning on a large GPU cluster. The convergence accuracy and throughput reflect great improvements.

3.	Training on low-bandwidth clusters is an important question and this paper solves it to some degree.

**Weaknesses:**

1.	In my opinion, the main insufficient of this paper is short of novelty. In fact, this paper is more like a technical report instead of an academic paper. The techniques of quantization and organizing the communication into a new topology are not new ideas. However, considering the overlap realization and trade-off between memory and communication are difficult and expensive, this paper is still significant for LLM community.

2.	Another problem is that this paper is lack of specific introduction to used quantization techniques and corresponding theoretical analysis. For example, how the quantization error affects the convergence rate and whether the error-feedback technique works.

**Questions:**

1.	What is the specific quantization methods in qwZ and qgZ?
2.	How to choose the quantization ratio during training and fine-tuning? I am interested in the inner effect of ratio on the convergence of different tasks.
3.	In the real low-bandwidth environment, does the trade-off between memory and communication still work without NVLink?

---

> ### Author Response · Authors · 2023-11-17
>
> Thank you for your insightful feedback. We have carefully considered your comments and addressed them in the following response.
>
> 1.	In my opinion, the main insufficient of this paper is short of novelty. In fact, this paper is more like a technical report instead of an academic paper. The techniques of quantization and organizing the communication into a new topology are not new ideas. However, considering the overlap realization and trade-off between memory and communication are difficult and expensive, this paper is still significant for LLM community.
>
> A: We agree that we are not the first to use quantization in communication and we appreciate the reviewer's acknowledgment of the significance of our work. In addition, we would like to note that ZeRO++'s novelty lies in its detailed application of quantization without impacting the convergence and improving system throughput. This is because, despite using state-of-the-art quantization to mitigate quantization error, a significant gap remains in terms of convergence due to numeric precision loss, and in throughput due to the computation cost involved in quantization and dequantization.   That is why we present a novel all-to-all communication primitive, a full set of CUDA kernels, and a carefully designed system to: i) maintain numeric precision even during communication in a quantized format; ii) minimize quantization costs using custom CUDA kernels; and iii) boost system throughput by concurrently performing intra-node and inter-node communication and hiding the quantization cost.
>
> 2.	Another problem is that this paper is lack of specific introduction to used quantization techniques and corresponding theoretical analysis. For example, how the quantization error affects the convergence rate and whether the error-feedback technique works.
>
> A:   Due to page limitations, details on the block-based quantization can be found in Appendix B.1, which we will clarify in the main text.
>
> Regarding theoretical convergence, one notable aspect is the unique design of ZeRO. In the ZeRO paradigm, there are two kinds of weights: the temporary weights, which are all-gathered from all GPUs in order to do forward/backward on a particular model layer and then discarded; and the local weight shard, which is used to populate the temporary weights, are permanent and will be updated at each optimizer step. ZeRO++ quantization only occurs when **we communicate the temporary weights**. Thus, the local shard of the weights will be free of the quantization error. For the gradients that are computed from the quantized temporary weights, there will be minor errors, yet fruitful research such as [1, 2, 3] prove that minor disturbance to gradients will not impact the overall convergence (the convergence rate follows O(1/sqrt{T}) for T iterations), which we have also empirically shown. An empirical evaluation can be found in Table 2 where the convergence gap increases as the quantization goes from 6 bits to 2 bits. We put the training curves of different bits in the appendix (Table 6). We also added the above discussion to the appendix to better discuss ZeRO++’s theoretical convergence.
>
> >[1] Gorbunov, Eduard, Filip Hanzely, and Peter Richtárik. "A unified theory of SGD: Variance reduction, sampling, quantization and coordinate descent." AISTATS 2020
> >
> >[2] Ramezani-Kebrya, Ali, et al. "NUQSGD: Provably communication-efficient data-parallel SGD via nonuniform quantization." The Journal of Machine Learning Research 22.1 (2021): 5074-5116.
> >
> >[3] Alistarh, Dan, et al. "QSGD: Communication-efficient SGD via gradient quantization and encoding." Advances in neural information processing systems 30 (2017).
>
>
> We view error-feedback as an orthogonal approach to ZeRO++ and we believe adding error-feedback techniques such as [4, 5] could help especially for 1-bit/2-bit training or for a smaller model, which is a part of our future work. We will have more discussion on this. Thank you for the suggestion.
>
> >[4] Tang, Hanlin, et al. "1-bit adam: Communication efficient large-scale training with adam’s convergence speed." International Conference on Machine Learning. PMLR, 2021.
> >
> >[5] Wu, Jiaxiang, et al. "Error compensated quantized SGD and its applications to large-scale distributed optimization." International Conference on Machine Learning. PMLR, 2018.
>
>
> 3.	What is the specific quantization methods in qwZ and qgZ?
>
> A: We use block-based quantization for both weights and gradients. The details of it can be found in Appendix B. Simply put, we pre-scale the tensors by small chunks before casting them to lower precision to mitigate the difference in numerical ranges and granularities.
>
> (To be continued)

---

> > ### Author Response · Authors · 2023-11-17
> >
> > 4.	How to choose the quantization ratio during training and fine-tuning? I am interested in the inner effect of ratio on the convergence of different tasks.
> >
> > A: The choice of quantization ratio is determined by the user's tolerance for the convergence gap. An empirical guide can be found in Table 2, where the convergence gap increases as quantization goes from 6 bits to 2 bits. The ZeRO++ system implementation also gives users the flexibility to quantize only part of the weights. By default, ZeRO++ skips quantization for bias, layernorm, and minor tensors as they commonly have a large impact on convergence and little impact on system throughput.
> >
> > 5.	In the real low-bandwidth environment, does the trade-off between memory and communication still work without NVLink?
> >
> > A: Consider an entry-level cluster without NVLink and high-bandwidth connections. For qwZ and qgZ, they offer a natural advantage because their communication reduction is uniform for both inter-node and intra-node communication, meaning that a reduction in bandwidth yields a greater benefit. For hpZ, it is still the case because when we consider the intra-node communication fall back to PCI, it is still much greater than Ethernet based inter-node communication so using intra-node communication to replace inter-node communication still makes sense. We have extended our evaluation to 10Gbps to show how ZeRO++ perform under low-bandwidth environment (see Appendix D).
> >
> >
> > We welcome any further questions or points of discussion.

---

### Author Response · Authors · 2023-11-17

Dear reviewers,

We would like to express our gratitude for all the constructive and insightful comments. Below, you will find our responses to each reviewer's comments. The manuscript has been revised accordingly, incorporating:

* Additional traces of convergence and theoretical convergence discussions. (Appendix D.1, D.2)
* Results from evaluations conducted in 10Gbps low-bandwidth scenarios, wherein ZeRO++ achieved up to a 4x speedup compared to the baseline. (Appendix D.3)

We are in the process of further refining the manuscript, with particular emphasis on the related work section and enhancing the clarity of our explanations.

If you have any follow-up question, please don't hesitate to ask!

---

### Meta-Review · Area_Chair_Sd1P · 2023-12-08

**Metareview:**

- This paper proposed a framework ZeRO++, which includes three main techniques to accelerate the training efficiency of large-scale models. The techniques include low-precision weight gathering, low-precision gradient aggregation, and communication topology refining.
- The strength of the paper is that the combination of the techniques shows impressive experimental results, which improve the communication efficiency a lot.
- The biggest concern of all the reviews holding the negative score is the lack of novelty. Each individual technique used in the submission is pointed out by the reviewers that may share some similarity with the previous mutual method.

**Justification For Why Not Higher Score:**

- The response does not address the concerns of the reviewers, especially for the review that holds the negative scores. For example, some reviewers may worry that the benefit of ZERO++ may be weakened when training FP8 modern H100. This is clearly correct, as both can reduce communication overhead, so using both simultaneously will inevitably result in a decrease in the gain of a single method. The combined use of parallel methods does not reduce the contribution of individual methods. The author emphasizes that ZERO++ is even better on H100, which may mislead reviewers.
- Regarding extra GPU memory, the author states that having almost no extra GPU memory is also misleading. ZERO++ has an additional buff of int4 during communication, and the author can claim that this buff will be released promptly (with a certain probability of not affecting peak memory), but cannot claim that there is no additional memory overhead.

**Justification For Why Not Lower Score:**

- Impressive experimental result.

---

### Decision · Program_Chairs · 2024-01-16

Accept (poster)